# Bacteriophages Biocontrol of Kiwifruit Bacterial Canker Caused by *Pseudomonas syringae* pv. *actinidiae* (Psa) in Two Seasons Under Field Conditions

**DOI:** 10.3390/antibiotics14101023

**Published:** 2025-10-14

**Authors:** Paulina Sanhueza, Natalia Riquelme, Marcela Leon, Javiera Gaete Morales, Camila Prince, M. Fernanda Flores, Carolina Yañez, Italo F. Cuneo, Roberto Bastías, Ximena Besoain

**Affiliations:** 1Escuela de Agronomía, Pontificia Universidad Católica de Valparaíso, Casilla 4D, Quillota 2260000, Chile; paulinasanhuezaq@gmail.com (P.S.); natalia.riquelme@pucv.cl (N.R.); javiera.morales@pucv.cl (J.G.M.); italo.cuneo@pucv.cl (I.F.C.); 2Millennium Nucleus Bioproducts, Genomic and Enviromental Microbiology (BioGEM), Avenida España 1680, Valparaíso 2390123, Chile; 3Laboratorio de Microbiología, Instituto de Biología, Pontificia Universidad Católica de Valparaíso, Av. Universidad #330, Curauma, Valparaíso 2340000, Chile; marcela.leon@pucv.cl (M.L.); camila.prince.casarotto@gmail.com (C.P.); carolina.yanez@pucv.cl (C.Y.); 4Agroadvance SpA, Camino a Melipilla 26200, Peñaflor 9750000, Chile; fflores@agroadvance.cl

**Keywords:** bacteriophage, *Pseudomonas syringae* pv. *actinidae*, Psa, Pss, biocontrol, Kiwifruit Bacterial Canker

## Abstract

**Background:** Since 2008, the kiwifruit industry has been significantly impacted by *Pseudomonas syringae* pv. *actinidiae* (Psa), the agent responsible for bacterial canker in kiwifruit. Existing treatments, such as copper-based compounds and antibiotics, have faced challenges related to resistance and soil contamination. Phage therapy is a promising and safe alternative for controlling this pathogen. This study aimed to evaluate the use of a mixture of four isolated and characterized bacteriophages as potential biocontrol agents against Psa. **Methods:** Trials were conducted at two locations in Chile, where Psa presence was reported during the 2019/2020 and 2020/2021 seasons, with a focus on the spring stages. Different formulations were tested each season to evaluate possible improvements in effectiveness. *Pseudomonas* spp. isolates obtained from epiphyte populations were characterized using morphological, biochemical (LOPAT), and molecular techniques. **Results:** Field trials demonstrated that the phage mixture effectively reduced the damage associated with Psa on kiwi leaves, resulting in a decrease in the *Pseudomonas* spp. bacterial load (42.9% for Peumo and 25% for Linares) at both locations during the first season trials. This decrease is associated with a reduction in the incidence and severity of the disease in kiwi plants in the Peumo orchard. In both seasons, bacteriophages reduce Psa symptoms in treated kiwi plants compared to untreated controls, at least at one location and evaluation. In both orchards during the first season, bacteriophages also outperformed copper- and antibiotic-based treatments used by farmers. Bacteriophage therapy is eco-friendly and safe for both applicators and consumers.

## 1. Introduction

*Pseudomonas syringae* pv. *actinidiae* (Psa) is the causal agent of bacterial canker in green-fleshed kiwi (*Actinidia deliciosa*) and yellow-fleshed kiwi (*A. chinensis*) and is currently the cause of severe economic losses in countries such as Italy, New Zealand, Portugal, Chile and South Korea [1,2,3]. Between 2008 and 2011, almost simultaneous re-emergence of the same aggressive pathogen occurred in all significant kiwi-producing areas worldwide; therefore, it was considered a pandemic disease [4,5]. In 2011, the pathogen was also officially reported for the first time in Chile, indicating that Psa control in kiwifruit is mandatory throughout the national territory [6,7].

*Pseudomonas syringae*, pv. *actinidiae* infection can cause symptoms such as necrotic spots with a yellow chlorotic halo on leaves during spring and summer. Wilting or withering of the flowers can occur in buds and flowers, resulting in the loss of future fruit. In early spring, abundant reddish-orange exudates are associated with cankers and wounds on the arms and trunks of infected plants. It is also possible to see cankers in the wood and branches or necrosis in the subcortical area without exudate [8]. Psa can penetrate natural openings (such as the stigma of flowers, stoma, lenticels) and wounds to start an infection. Subsequently, it spreads rapidly in host tissues, causing severe symptoms, crop losses, and, eventually, plant death [3,7]. Disease development is associated with specific environmental conditions [3,9,10], primarily in the spring and autumn. Temperatures between 10 °C and 25 °C, water availability, rain, and wind facilitate pathogen dispersal [4,7,11]. These conditions greatly favor the multiplication of the bacterium, allowing it to spread systemically from the leaf to young shoots. During summer, very high temperatures can reduce the multiplication and dispersal of bacteria [4].

Currently, five different biovars are recognized within Psa (biovars 1, 2, 3, 5, and 6). Biovar 4 was previously considered a less virulent variant of Psa; however, it is now classified as part of the pathovar *actinidifoliorum* (Psaf) [7]. In Chile, to date, only biovar 3 has been reported [12], corresponding to the most virulent biovar of this pathovar, which is also responsible for the most recent outbreaks of Psa worldwide, including those in Europe, New Zealand, and, more recently, Japan and Korea [7].

The global outbreak of Psa has prompted research to develop effective strategies for containing this pathogen and minimizing economic losses in the kiwifruit industry. Currently, the control methods against Psa include adopting preventive practices to reduce the proliferation of outbreaks, such as constant monitoring and early detection, disposal of infected material, disinfection of tools, and the use of biological control agents (BCAs) and elicitors. However, most importantly, the frequent spraying of orchards with copper-based compounds, particularly cuprous oxide (Cu_2_O), and antibiotic formulations containing streptomycin [13], has had limited success [4,14,15]. The application of these chemical compounds favors the emergence of Psa strains resistant to copper and antibiotics [7,16], which have been reported to possess *copA* and *copB* copper resistance genes and other antibiotic resistance genes [17,18,19]. Authorized treatments for this disease are still scarce, their efficacy is limited, and most are not environmentally friendly [20,21,22]. Evidence of phytotoxicity caused by the continued use of Cu, combined with European regulations restricting these agrochemicals, highlights the need for new control strategies.

One alternative is to use bacteriophages as a biocontrol strategy to eliminate bacteria. Phages are environmentally friendly, can be highly specific [20], and, unlike broad-spectrum antimicrobials, they do not affect the normal microbiota of plants [23,24,25]. Furthermore, bacteriophage treatments have no harmful effects on humans, animals, or plants [26,27]. Other advantages of phages over antibiotics, such as bactericides and copper-based chemicals, include their natural ubiquity in the biosphere, self-replication within the bacterial host, and therefore accumulation where they are most needed [23,28,29,30,31,32]. These characteristics make phages a promising and natural antibacterial method for controlling Kiwifruit Bacterial Canker [1,25].

The use of bacteriophages to control pathogenic bacteria associated with agriculture has been proposed previously. A review published in 2017 [33] summarizes different studies on the use of bacteriophages against *Pectobacterium carotovorum* spp., *Dickeya solani*, *Ralstonia solanacearum*, and *Xanthomonas campestris* pv. *versicatoria*, *Xylella fastidiosa*, *Pseudomonas syringae* pv. *porri*, and *Pseudomonas tolaasii*. A study in New Zealand [1], based on in vitro assays, evaluated the potential of phages for Psa control and showed that their phages could infect different strains of Psa isolated from different countries. Another study in Portugal [34] reported that phage ф6 could reduce the bacterial concentration of Psa in ex vivo tests on kiwifruit leaves under laboratory conditions. Other groups have reported the isolation and characterization of phages against Psa in Italy [35,36,37], Korea [38,39], and China [24,25,40]. A recent review summarizes different experiences with the use of bacteriophages as biocontrol agents for *P. syringae* [41].

In Chile, a recent report [42] isolated and characterized phages against Psa. A group of selected phages were evaluated for their effectiveness under laboratory and greenhouse conditions, demonstrating their potential as biocontrol agents. Despite numerous reports on the potential use of bacteriophages to control bacterial cankers in kiwi plants, none of them have demonstrated their efficacy under field conditions. Therefore, it is imperative to gain a deeper understanding of the efficacy of phages in controlling Psa under natural conditions and thus transfer this technology to kiwifruit orchards [34].

In this study, we evaluated the effects of foliar application of a mixture of specific bacteriophages against *Pseudomonas syringae* pv. *actinidiae* to control the disease in two productive kiwifruit orchards in two different seasons and compared the effectiveness of phage treatment with regular conventional bactericide treatments.

## 2. Results

### 2.1. First Season 2019/2020

#### Bacteriophage Formulations’ Effect on Kiwifruit Bacterial Canker

During the first trial season (2019/2020), the foliar damage index (DI) was evaluated for kiwifruit plants in two fields located in Peumo and Linares. During spring, data were recorded on three occasions: 30, 45, and 90 days after the first application (DAA) (Figure 1A–F).

In Peumo, the effect of farmer management (T1) was not different from that of the control (T0) at 30, 45, and 90 DAA (Figure 1A–C). In the trial conducted in Linares, the data showed consistently worse effects of farmer management (Figure 1D–F). This treatment increased the damage index significantly in comparison to the control treatment (T0).

In Peumo during the second evaluation (Figure 1B), all three phage formulations (T2, T3, and T4) showed statistical differences from the farmer’s management (T1), and (T2, T4) were also statistically different from the control (T0), being efficient in controlling the damage caused by the disease. In Linares (Figure 1E), the phage formulation (T3) was also different from the control treatment (T0) and the farmer’s treatment.

### 2.2. Second Season 2020/2021

#### 2.2.1. Bacteriophage Formulations Effect in Kiwifruit Bacterial Canker

During the second season trials (2020/2021), the effect of bacteriophage formulation in Peumo, statistical differences between treatments were observed at 90 DAA. Although no significant differences were observed between any treatment and the control treatment (T0), significant differences were noted between the phage treatment (T3) and both the conventional treatment (T1) and the milk treatment (T2), but no differences between the control treatment. These results reinforce the potential of phages to control Psa infections, although they also indicate that further development is needed for this formulation.

In Linares was observed in the second and third evaluations (Figure 2E,F), that plants sprayed with phage-based treatments in LB medium (phage formulation 1) (T3) presented the lowest DI; however, it was at 45 DAA when statistical differences were observed in comparison with the control treatment (T0), indicating the efficacy of the treatment in reducing symptomatology. In Linares’s location, at 90 DAA (Figure 2F), the use of milk (T2) and its use as an excipient for phage formulation (T4) show a negative impact on the DI measures, compared to phage treatment (T3). Although phage treatment (T3) did not show statistically significant differences compared to the control treatment (T0), it consistently exhibited the lowest damage value.

#### 2.2.2. Epiphytic Bacterial Populations, LOPAT, and Molecular Identification

Only in the first season, epiphytic bacterial populations isolation was performed (Figure 3). Samples were collected to isolate and identify the bacteria present on the kiwi leaves before treatment applications.

The epiphytic bacterial load was high in the control treatment (T0) group and in the early stages of treatment, ranging from 10^6^ to 10^8^ CFU/mL (Figure 3A,D). The bacterial load decrease by up to 43% after 60 days of treatment with phage formulations 1 and 2 (Figure 3), corresponding with a decrease in foliar damage observed in the first season of evaluation in Peumo (Figure 1B,C).

Representative bacteria were recovered from epiphytic populations, including *P. fluorescens*, other *Pseudomonas* species, and other genera (Table 1 and Table 2). These organisms could exert natural control of Psa and Pss bacteria and may exhibit resistance to copper or antibiotic treatments. The effect of these bacterial populations should be evaluated in the future.

Gram-negative and fluorescent isolates were subjected to the LOPAT test, classifying bacteria as *Pseudomonas* spp. Based on the LOPAT profiles (Levan production (L), presence of oxidase (O), soft rot in potato (P), presence of arginine dihydrolase (A), and hypersensitivity reaction in tobacco leaves (T)), the following species were identified: *Pseudomonas fluorescens* (6 isolates) (Table 1), *Pseudomonas syringae* (13 isolates), *Pseudomonas viridiflava* (1 isolate), and *Pseudomonas marginalis* (1 isolate). Three isolates did not match with the LOPAT profile [43]. Subsequently, a subset of 17 isolates was sequenced to confirm their identity. Finally, specific primers were used in the PCR assays to identify *P. syringae* pv. *actinidiae* (Psa) and *P. syringae* pv. *syringae* (Pss), and *P. viridiflava* (Pv) (Table 1 and Table 2).

#### 2.2.3. Pathogenicity Test

Thirteen isolates identified as *Pseudomonas syringae* were selected to corroborate their pathogenicity in excised kiwifruit leaves. Of the total isolates analyzed (*n* = 13) (Appendix A), ten of them were pathogenic on excised kiwifruit leaves (76.9%) after 7 days post-inoculation, causing necrotic lesions on inoculated leaves, and 15 days after inoculation, all the *P. syringae* were pathogenic in leaves. The isolate, identified as *P. viridiflava*, did not show any lesions. The first symptoms appeared 5–7 days after inoculation. Infected leaves showed 3–7 lesions, which became red brown, with curling of the leaf lamina. The bacteria were re-isolated from these lesions and identified as *Pseudomonas* according to their morphology and fluorescence under UV light. Leaves inoculated with water showed no symptoms.

## 3. Discussion

The relationship between lytic bacteriophages and pathogenic bacteria has been studied for a long time to control pathogen populations in different fields, primarily under laboratory or greenhouse conditions [21,44,45,46,47,48,49,50,51,52]. Therefore, it is necessary to conduct studies that help collect data on the use of phage cocktails in the field.

Applying phage formulations under field conditions has been challenging due to various environmental factors that influence phage cocktail performance, such as temperature, pH, and UV radiation [53]. Numerous bacteriophages have demonstrated significant potential under laboratory conditions [1,35,38,42,54,55,56,57]; however, not all exhibit comparable efficacy under field conditions, which has led to the development and evaluation of different types of formulations that aim to extend phage viability under field conditions [30]. The results obtained in this study align with previous analyses that guided the formulation of tested phage cocktails [42,57]. Specifically, bacteriophages CHF1, CHF7, CHF19, and CHF21 possess characteristics that enable the effective control of *Pseudomonas syringae* pv. *actinidiae* (Psa) under in vitro, in vivo, and greenhouse-controlled conditions [42]. Our findings complement previous work on *P. syringae* pv. *actinidae*, where phenotypic and genetic variation were linked to pathogenicity and potential differences in phage susceptibility [12]. Consistent with phage selection principles [1], the in vitro and ex vivo efficacy of phage ф6 [34] and plant-level prophylactic models [37] support the feasibility of phage-mediated biocontrol and highlight host range and application timing as critical determinants.

It is worth noting that Pss isolates may not be controlled by the phages used in this study for the control of Psa; phages are generally very specific. However, a study conducted by Amarillas et al. [58] (2020) reports that phages specific to Pss also have some, albeit erratic, control over *P.s.* pv. *tomato* and, on the other hand, do not control *P.s.* pv. *phaseolicola*. This aspect warrants further study by our group.

Trials conducted over both seasons highlighted one of the main challenges in controlling this disease in Chile and other kiwifruit-producing countries: the resistance of pathogens to copper-based or antibiotic products. Copper compounds are among the most widely used agents for managing *Pseudomonas syringae* pv. *actinidiae* (Psa), but their efficacy remains limited because they mainly provide surface protection and do not prevent internal colonization by pathogens [19]. In this study, conventional management practices exhibited minimal or no control effects. This result is probably associated with the appearance of resistance to copper or antibiotics in Psa [17,18,19].

*Pseudomonas* species such as Pss and Psa can be resistant or less sensitive to copper and antibiotics, which may explain the lack of effectiveness of treatments carried out by farmers. Other authors in Chile have not studied this aspect; however, reports by Cordova et al. (2022) [59] confirm the presence of *P. s*. pv. *tomato* isolates resistant to copper and streptomycin, although not to oxytetracycline. In this study, they also evaluate the resistance of other isolated *Pseudomonas* species, such as *P. fluorescens*, *P. marginalis*, and *P. viridiflava*, where isolates of these species were found to be highly resistant to copper and streptomycin.

The trials were conducted in orchards where Psa had been previously detected; however, due to restrictions imposed by regulatory agencies, experimental Psa infection of these crops was not possible. In this sense, the detection of Psa in samples obtained from control treatments confirms the presence of this bacterium in these fields.

Among these bacterial isolates, it is noteworthy that both Psa and Pss demonstrated virulence in our assays. Although Psa is recognized as the causal agent of bacterial cankers in kiwifruit, similar symptoms have also been associated with Pss, including leaf wilt, cane blight, dried bark, brown necrotic tissue, and rusty red exudate [60,61,62]. Furthermore, other studies have reported frequent co-occurrence of *P. syringae pv. syringae* and *P. viridiflava* in kiwifruit; however, their interactions remain poorly understood [63,64,65].

Considering this, it would be desirable to develop specific phages for *Pss* to increase the reduction in kiwi foliar symptoms, or polyvalent cocktails able to combat different phytopathogens associated with kiwifruit trees.

Nevertheless, phage therapy developed for Kiwifruit Bacterial Canker was effective in reducing the disease, and this is the first report of its efficacy under field conditions at two different locations and in two different seasons.

## 4. Materials and Methods

### 4.1. Fields Experiments

The trials were simultaneously implemented in two locations: the first field in Peumo, O’Higgins Region, Chile, and the second in Yerbas Buenas, Linares, Maule Region, Chile. The trials were conducted over two seasons (2019–2020 and 2020–2021) during the budding and flowering (spring) stages, when there was a higher risk of infection. Due to regulatory restrictions of the Agriculture and Livestock Service of Chile, Servicio Agrícola y Ganadero (SAG), which declared “Mandatory control of Psa pest in kiwi species” in all territory from 2011 to 2020 [66], the orchards used for field trials were naturally infected with Psa; however, both fields had positive records of Psa presence. The trials were performed in blocks of commercial kiwifruit orchards (*A. deliciosa* var. *Hayward*) planted 5 × 2 m^2^.

### 4.2. Bacteriophage Formulation

Based on the results of previous research [12,42], the phages CHF1 (GeneBank MN729595), CHF7 (GeneBank MN729596), CHF19 (GeneBank MN729597), and CHF21 (GeneBank MN729598) were selected for this study (Table 3). These phages effectively controlled the bacterium under laboratory and greenhouse-controlled conditions [42]. Phage propagation was performed by mixing Psa strain 889 [12] in the exponential phase with each phage separately (MOI 0.5) and incubating the mixture at 25 °C with constant shaking for 5 h. To remove debris, lysates were centrifuged at 10,000 rpm for 20 min at 4 °C, and then filtered through a 0.22-µm-pore-size filter and stored at 4 °C. The phage lysates were precipitated with 10% (*w*/*v*) polyethylene glycol (PEG 8000) and 1 M NaCl overnight at 4 °C and then centrifuged (14,000 rpm for 10 min at 4 °C). The pellet was resuspended in magnesium buffer (10 mM MgSO_4_), and 1% chloroform was added. After centrifugation (14,000 rpm for 10 min at 4 °C), the upper aqueous phase was stored at 4 °C. Phage titer was determined using a double-agar plate assay [67]. The four phages were mixed in equal proportions for the application, and excipients were added when specified. Different formulations were used in each season to evaluate potential increases in effectiveness. The composition of each formulation assayed in each season is presented in Table 4 and Table 5.

### 4.3. First Season 2019–2020 Field Trials

Treatments were distributed following a design of randomized complete blocks, using three plants as experimental units and four blocks as replicates; 12 plants for each treatment were assayed. In addition, between each treatment, one plant was left untreated and used as a border barrier to isolate different treatments and replicates. Foliar applications of the five treatments described in Table 4 were performed using a high-pressure cart sprayer at 500 psi and 1.5 bar power (EISEN^®^, Eisen Electric Corporation, Lansing, MI, USA). The phages were sprayed at two different concentrations quantified as plaque-forming units per mL (PFU/mL): Formulation 1 with 1 × 10^6^ (PFU/mL) and Formulation 2 with 1 × 10^7^ (PFU/mL), at a rate of 1000 L/ha. The phage mix treatments were applied weekly for two months (eight applications) during the spring.

For the conventional treatment associated with the farmer’s traditional management, copper sulfate and a streptomycin-based bactericide were applied interspersed. In spring, eight applications were made during budding, flowering, and fruit set, with a wetting of 1000 L/ha and a 7-to 10-day frequency, depending on the phenological stage. The control plants received only a water spray.

### 4.4. Second Season 2020/2021 Field Trials

The trials were conducted by replicating the previous season’s model by incorporating a new formulation into the phage-based treatments. All the formulations used during the second season are listed in Table 5. The phage mix treatments were applied weekly for two months (eight applications) during the spring season. The phages were sprayed at 1 × 10^7^ (PFU/mL). The management of conventional farmers was also based on copper sulfate and streptomycin-based bactericides in season 1. All treatments were performed with a wetting rate of 1000 L/ha. Eight applications were made in the spring during budding, flowering, and fruit sets, with a seven-day interval between each. The control plants were treated with water only.

### 4.5. Effect of Treatments Under Kiwifruit Bacterial Canker in Field Trials

The effectiveness of the different treatments was evaluated by monitoring the symptoms of the Kiwifruit Bacterial Canker at 30, 45, and 90 days after the first application. Forty leaves were randomly selected from each tree and evaluated for each treatment. The DI of each leaf was evaluated using a damage scale for Psa proposed by Flores et al. [42] with the following parameters 0: healthy leaf; 1: 1–4% of leaf area affected; 2: 5–10% of leaf area affected with single spots and few merged spots; 3: 11–30% of leaf area affected with merged spots; 4: 31–49% of leaf area affected, merged spots covering veins and increasing in size; 5: >50% of leaf area affected (Appendix A). Damage index (DI) was calculated for each treatment using the following formula:DI% = ∑((n*v)/(V*N))100)
where n = number of leaves per degree of attack; v = degree of attack (0,1,2,3,4,5); N = Maximum range of the scale; V = Total number of leaves evaluated. Statistical analysis was performed using one-way ANOVA and the Kruskal–Wallis test (*p* ≤ 0.05).

### 4.6. Epiphytic Bacterial Populations in First Season

Bacterial epiphyte populations were detected following the methodology described by Purahong et al. [65]. Ten leaves were randomly selected and collected per sampling unit (*n* = 3 trees) for a total of 40 leaves per control treatment (ten per replicate). Leaves were taken before the treatment’s application and 30 and 60 days after the first application, coinciding with the last application. The collected samples were transported from the field to the Laboratory of Microbiology (PUCV) for processing in tight plastic bags and inside a plastic cooler container at 5 °C. The 10 leaves sampled per replicate were pooled for processing. Each leaf sample was washed with 15 mL of sterile MgSO_4_ 10 mM buffer solution and 0.1% Tween 20™; washing was carried out for 30 min with gentle agitation (100 rpm). Serial dilution and plate count (on King’s medium B) of the wash solution were used to quantify the bacterial population. The plates were then incubated for 48 h at 25 °C. Subsequently, individual colonies were isolated, lyophilized, and stored at −20 °C for subsequent identification. This isolation was only performed during the first year of the trials (2019) because of the COVID-19 pandemic, and epiphytic population analysis could not be achieved in the second season.

### 4.7. Detection and Identification of Pseudomonas spp.

#### 4.7.1. Morphological, Biochemical, and Molecular Test

Various methods have been used to identify *Pseudomonas* spp. The purified and lyophilized isolates were cultured on King’s B medium to verify their ability to produce fluorescence under UV light [68]. Based on Lelliot and Stead [43], the LOPAT test (Levan production (L), presence of oxidase (O), soft rot in potato (P), presence of arginine dihydrolase (A), and hypersensitivity reaction in tobacco leaves (T)) was performed on a representative number of isolates of the *Pseudomonas* genus. These tests identified *P. syringae* (possibly Psa and Pss) with a LOPAT profile (+---+). At the same time, *P. viridiflava* presents a LOPAT profile (--+-+).

#### 4.7.2. Bacterial DNA Extraction and Molecular Identification

Seventeen isolates (58.6% of the isolates) were cultivated for DNA extraction. Bacterial DNA was extracted from individual colonies grown for 48 h at 25 °C in King’s medium B using a Wizard^®^ Genomic DNA Purification Kit (Promega, Madison, WI, USA), according to the manufacturer’s instructions. The DNA was used to sequence the 16S rDNA using the universal primers 27F and 907R [69,70]. PCR was carried out in a 23 μL reaction mixture containing 1 μL of genomic DNA (<1 ng), 12.5 μL of SapphireAMP^®^ Fast PCR Master Mix (2X) (Takara, Tokyo, Japan), and 0.4 μM of each primer. The amplification program consisted of an initial denaturation at 94 °C for 1 min, followed by 30 cycles of 98 °C for 5 s, 55 °C for 5 s, and 72 °C for 10 s. PCR products were separated by agarose gel electrophoresis (1.5% agarose in 1× TAE buffer) and stained with GelRed (Biotium, Fremont, CA, USA), and the bands were visualized under UV light (excitation at 256 nm). PCR products were sent to Macogen (Seoul, Republic of Korea) for sequencing, and the sequences were processed using the Geneious R10 program. The BLAST 2.17.0 tool was used to establish the identities of the sequenced isolates. PCR using specific primers for *P. s.* pv. *actinidae* were obtained through PCR with PsaF- and PsaR-specific primers [71], B1 and B2 specific primers for the *syrB* gene of Pss [72], and PsV-F and PsV-R primers specific for *P. viridiflava* [73]. In all cases, the PCR reactions were performed in a SureCycler 8800 thermal cycler (Agilent Technologies, Santa Clara, CA, USA) using Sapphire Amp Fast PCR Master Mix (Takara Bio, Shiga, Japan). Each reaction was carried out in a final volume of 25 μL, containing 2 μL of genomic DNA (<1 ng), 11.85 μL of SapphireAMP^®^ Master Mix (2X) (Takara, Tokyo, Japan), and 0.5 μM of each primer. The amplification program consisted of 35 cycles of denaturation at 94 °C for 1.5 min, annealing at 60 °C for 1.5 min, and extension at 72 °C for 3 min, followed by a final elongation at 72 °C for 10 min. PCR products were separated by agarose gel electrophoresis (1.5% agarose in 1× TAE buffer) and stained with GelRed (Biotium), and the bands were visualized under UV light (excitation at 256 nm).

#### 4.7.3. Pathogenicity Test

Based on the protocol of Alimi et al. [73], a pathogenicity test was carried out on the leaves of *A. deliciosa* Hayward. Kiwi leaves were obtained from a healthy nursery plant, 20 mL of bacterial suspensions of each isolate adjusted to 1 × 10^7^ CFU/mL were sprayed on the leaves using three leaves per bacterium as replicates. Controls were sprayed with sterile distilled water. Treated leaves were kept at 20 °C in humid plastic trays for 15 days. Symptoms were observed 7 and 15 days after inoculation. Bacteria were re-isolated after necrotic spots appeared, and colonies were identified morphologically by Gram staining and fluorescence using UVA (320 nm), fulfilling Koch’s postulates (Appendix A).

### 4.8. Data Analysis

Analysis of variance (ANOVA) was conducted to evaluate the significance of the treatment effects. Before the analysis, assumptions of normality and homogeneity of variances were verified using the Shapiro–Wilk and Levene’s tests, respectively, implemented in the InfoStat software version 2017 [74]. Post hoc comparisons among means were performed using Tukey’s HSD test at the 5% significance level. Graphical representations were generated using the RStudio Posit Team software 2025. Error bars in the figures indicate standard deviations, and different letters above the bars denote statistically significant differences among treatments.

## 5. Conclusions

Field trials conducted over two consecutive seasons in Linares and Peumo support the efficacy and sustainability of bacteriophage-based formulations for controlling *Pseudomonas syringae* pv. *actinidiae* (Psa) in kiwifruit. The results show that bacteriophage formulations consistently reduced the damage index (DI%), even surpassing copper treatments, at least in one evaluation in three of the four trials conducted.

Identification tests performed on the isolates obtained confirmed the presence of strains of the *Pseudomonas syringae* complex, including *P. syringae* pv. *actinidiae*, *P. syringae* pv. *syringae*, and *P. viridiflava*. These findings validated the natural presence of pathogens in the field, supporting the appropriateness of the trial conditions. Pathogenicity tests confirmed that 76.9% of *P. syringae* isolates were virulent, generating necrotic lesions on plants 5–7 days after inoculation.

Together, these results support the use of phage-based formulations as specific, effective, and environmentally friendly tools for the integrated management of Kiwifruit Bacterial Canker.

## 6. Patent

KIWIPHAGE formulation, based on four bacteriophages, was entered into the INAPI registry for intellectual protection of this technology. Chilean Patent Application No. 2021-1727. The registration number is 68.232, under the title “Composición antimicrobiana para controlar el cancro bacteriano del kiwi”.

## Figures and Tables

**Figure 1 antibiotics-14-01023-f001:**
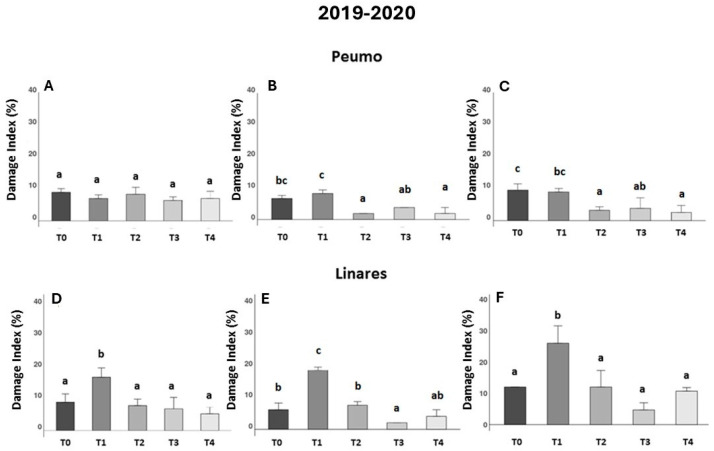
Effectiveness of treatments against bacterial canker development in two kiwifruit fields during the first season (2019–2020). Evaluated using the Damage Index (DI), corresponding to the percentage of leaf area in the presence of necrotic spots (*n* = 10). Data were recorded 30, 45, and 90 days after treatments were applied in Peumo (**A**–**C**) and Linares (**D**–**F**). (T0) Control only water; (T1) Farmer’s management; (T2) Phage formulation 1; (T3) Phage formulation 2; (T4) Farmer’s management intercalated with phage formulation 2. Error bars represent standard deviations. Different letters above the bars indicate statistical differences among treatments according to Turkey’s HSD test (*p* < 0.05).

**Figure 2 antibiotics-14-01023-f002:**
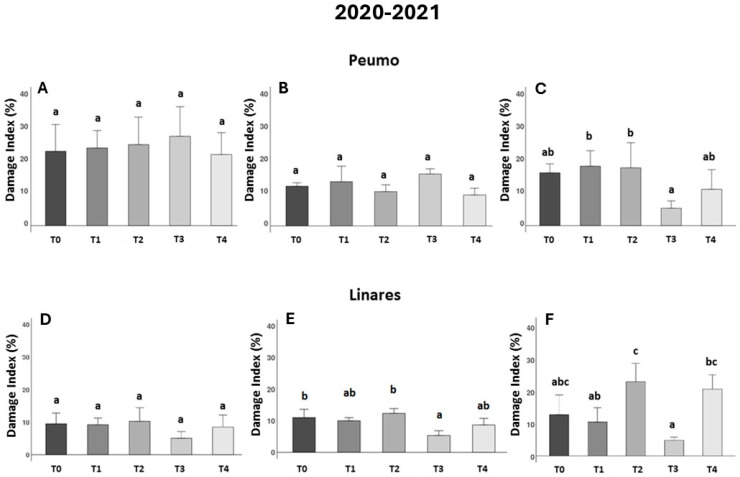
Effectiveness of treatments against bacterial canker development in two kiwifruit fields during the second season (2020–2021). The Damage Index (DI) corresponds to the leaf area percentage with necrotic spots (*n* = 10). Data were recorded 30, 45, and 90 days after treatments were applied in Peumo (**A**–**C**) and Linares (**D**–**F**). (T0) Control only water; (T1) Farmer’s management; (T2) Only milk; (T3) Phage formulation 1; (T4) Phage formulation 2 (including milk). Error bars represent standard deviations. Different letters above the bars indicate statistical differences among treatments according to Turkey’s HSD test (*p* < 0.05).

**Figure 3 antibiotics-14-01023-f003:**
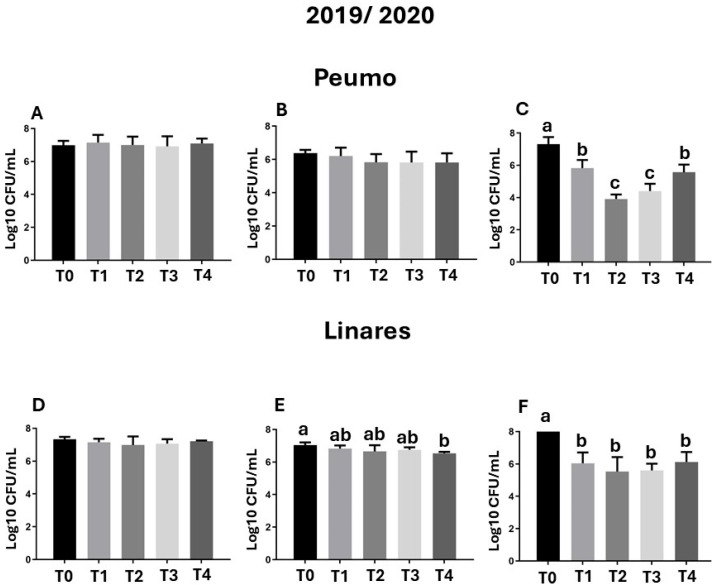
Epiphytic bacterial populations. Samples were evaluated before treatment application, 30 days after treatment application (DAA), and 60 DAA in Peumo (**A**–**C**) and Linares (**D**–**F**). The total bacterial population (CFU/mL) per plate was quantified. Each sample pool was processed individually by plate dilution of a homogenate prepared from 10 leaves. (T0) Control only water; (T1) Farmer’s management; (T2) Phage formulation 1; (T3) Phage formulation 2; (T4) Farmer’s management intercalated with phage formulation 2. Error bars represent standard deviations. Different letters above the bars indicate statistical differences among treatments according to Turkey’s HSD test (*p* < 0.05).

**Table 1 antibiotics-14-01023-t001:** Results of LOPAT [43] performed on the representative of the group of fluorescent *Pseudomonas* recovered during the first season of Peumo and Linares trials.

Isolates	Orchard	DAA	Levan Type Colonies	Oxidasa Reaction	Potato Root	Arginine Dihydrolase	Tabacco Hyper-Sensitivity	Species
FF4	Peumo	0	+	+	−	+	−	*P. fluorescens*
FF6	Peumo	30	+	+	−	+	−	*P. fluorescens*
FF2	Peumo	30	+	+	−	+	−	*P. fluorescens*
3C	Peumo	60	+	−	−	−	+	*P. syringae*
BP2	Peumo	30	+	−	−	−	+	*P. syringae*
BP4	Peumo	30	+	−	−	−	+	*P. syringae*
BP10	Peumo	0	+	−	−	−	+	*P. syringae*
F1	Peumo	0	+	−	−	−	+	*P. syringae*
BM1	Peumo	60	+	−	−	−	+	*P. syringae*
BP7	Peumo	0	+	−	−	−	+	*P. syringae*
BP1	Peumo	60	+	−	−	−	+	*P. syringae*
AM3	Peumo	60	+	−	−	−	+	*P. syringae*
BP12	Peumo	0	+	−	−	−	+	*P. syringae*
BP8	Peumo	60	−	−	+	−	+	*P. viridiflava*
FGL5	Linares	30	+	+	−	+	−	*P. fluorescens*
FGL10	Linares	30	+	+	−	+	−	*P. fluorescens*
FPL8	Linares	60	−	+	−	+	−	*P. fluorescens*
FPL7	Linares	60	+	−	−	−	+	*P. syringae*
FPL11	Linares	30	+	−	−	−	+	*P. syringae*
BML4	Linares	60	+	−	−	−	−	*P. putida **
BML2	Linares	60	−	+	−	+	−	*P. putida **
FPL5	Linares	60	+	+	+	−	−	*P. marginalis*
FPL3	Linares	60	−	−	−	−	−	*P. umsongensis **

* Isolates taxonomically assigned exclusively based on 16S rRNA gene sequence analysis.

**Table 2 antibiotics-14-01023-t002:** Molecular and phenotypic characterization of bacterial isolates from kiwifruit leaves based on LOPAT test, species-specific PCR primers, 16S rRNA gene sequencing, and *syrB* gene analysis.

Code Isolates	Species by LOPAT	Primers Psa/Pv PCR	Gen rDNA 16s PCR	Ident (%)	Acc. Number Reference	Gen *syrB PCR*	Acc. Number Reference	Ident (%)	Species
FGL5	*P. fluorescens*		*P. fluorescens*	100	*MN511732*				*P. fluorescens*
FF2	*P. fluorescens*		*P. fluorescens*	100	*MN511732*				*P. fluorescens*
FGL10	*P. fluorescens*		*P. fluorescens*	100	*MN511732.1*				*P. fluorescens*
FPL7	*P. syringae*	*P.s. actinidia*							*Psa*
3C	*P. syringae*	*P.s. actinidia*							*Psa*
FPL11	*P. syringae*	*P.s. actinidia*							*Psa*
BP2	*P. syringae*		*P. syringae*	99.8	*MK388374.1*	*P.s. syringae*	*MK453199*	93.2	*Pss*
BP4	*P. syringae*		*P. syringae*	100	*CP047267*	*P.s. syringae*	*MK453199*	93.3	*Pss*
BP10	*P. syringae*		*P. syringae*	100	*CP047267*	*P.s. syringae*	*MK453199*	93.1	*Pss*
F1	*P. syringae*		*P. syringae*	100	*CP047267*	*P.s. syringae*	*MK453199*	93.8	*Pss*
BM1	*P. syringae*		*P. syringae*	100	*KC816628.1*	*P.s. syringae*	*MK453199*	93.6	*Pss*
BP7	*P. syringae*		*P. syringae*	100	*MK637897*	*P.s. syringae*	*MK453199*	93.2	*Pss*
BP1	*P. syringae*		*P. syringae*	100	*LC508793.1*	*P.s. syringae*	*MK453199*	93.2	*Pss*
AM3	*P. syringae*		*P. syringae*	100	*MK637897*	*P.s. syringae*	*MK453199*	93.2	*Pss*
BP12	*P. syringae*		*P. syringae*	96.86	*KF681132*				*P. syringae*
BML4	*n.i.*		*P. putida*	99.87	*KM187292*				*P. putida*
FPL5	*P. marginalis*								*P. marginalis*
FPL3	*n.i.*		*P. umsongensis*	99.62	*CP044409*				*P. umsongensis*
BML2	*n.i.*		*P. putida*	100	*KM187292*				*P. putida*
BP8	*P. viridiflava*	*P. vidiriflava*	*-*	99.63	*MN989115*				*P. viridiflava*
BPL12	*n.c.*		*Acinetobacter guillouiae*	100	*MT322954*				*A. guillouiae*
BPL7	*n.c.*		*A. guillouiae*	99.89	*MG517433*				*A. guillouiae*
BPL10	*n.c.*		*A. guillouiae*	100	*MH144279*				*A. guillouiae*
BP5	*n.c.*		*A. guillouiae*	99.76	*MT322954*				*A. guillouiae*
AM2	*n.c.*		*Curtobacterium flaccumfaciens*	99.8	*MN826580*				*C. flaccumfaciens*
AML2	*n.c.*		*C. flaccumfaciens*	100	*MN826580*				*C. flaccumfaciens*

*n.i.*: not identified; LOPAT analysis was performed but did not yield a conclusive outcome for species identification. *n.c.*: LOPAT analysis not conducted; isolates were not fluorescent under UV light when grown on MBK medium.

**Table 3 antibiotics-14-01023-t003:** Genomic characteristics of Mix Chilean Psa phages used in treatments [42].

Phage	Genome Size (pb)	%GC	ORFs ^a^	Genus	Identity (%) phiPSA2 ^b^
CHF1	40.999	57.3	49	T7-like	94.0%
CHF7	40.557	57.4	48	T7-like	96.4%
CHF19	40.882	57.3	48	T7-like	93.2%
CHF21	40.557	57.4	48	T7-like	93.8%

^a^. Number of ORFs found in each genome. ^b^. Psa phage genomes were aligned with phiPsa2 using Mauve.

**Table 4 antibiotics-14-01023-t004:** Details of foliar spray treatments in kiwifruit orchards in Peumo (O’Higgins Region) and Linares (Maule Region), Chile, during the first trial season (2019–2020).

Code	Treatment	Composition	Rate
T0	Control	Water	-
T1	Farmer’s management based on intercalary bactericides	Copper sulfate; Streptomycin (25%), Oxytetracycline (3.2%), and co-formulants	60 g/100 L (*m*/*v*)
T2	Phage formulation 1 *	A mixture of four lytic bacteriophage	1 × 10^6^ (PFU/mL)
T3	Phage formulation 2 *	A mixture of four lytic bacteriophages	1 × 10^7^ (PFU/mL)
T4	Intercalary application of bactericides and phage Formulation 2	Copper sulfate, a mixture of four lytic bacteriophage	60 g/100 L; 1 × 10^7^ (PFU/mL)

* Phage Formulation Mix compounds about four phages: CHF1 (MN729595); CHF7(MN729596); CHF19(MN729597); CHF21(MN729598).

**Table 5 antibiotics-14-01023-t005:** Details of foliar spray treatments in kiwifruit orchards in Peumo (O’Higgins Region) and Linares (Maule Region), Chile, during the second trial season (2020–2021).

Code	Treatment	Composition	Rate
T0	Control	Water	-
T1	Farmer’s management based on intercalary bactericides	Copper sulfate; Streptomycin (25%), Oxytetracycline (3.2), and co-formulants	60 g/100 L (*m*/*v*)
T2	Only Milk	low-fat milk 5% (*m*/*v*)	5 kg/100 L (*m*/*v*)
T3	Phage formulation 1 *	Mixture of four lytic bacteriophages and LB (Lysogeny Broth) Medium	1 × 10^7^ (PFU/mL)
T4	Phage formulation 2 *	Mixture of four lytic bacteriophages and low-fat milk 5% (*m*/*v*)	1 × 10^7^ (PFU/mL); 5 kg/100 L (*m*/*v*)

* Phage Formulation Mix compounds about four phages: CHF1 (MN729595); CHF7(MN729596); CHF19(MN729597); CHF21(MN729598).

## Data Availability

The data presented in this study are available on request from the corresponding author.

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
