# Peer review of "Bacteriophages Biocontrol of Kiwifruit Bacterial Canker Caused by Pseudomonas syringae pv. actinidiae (Psa) in Two Seasons Under Field Conditions"

_antibiotics, 2025, doi:10.3390/antibiotics14101023_

Round 1

Reviewer 1 Report

Comments and Suggestions for Authors

Abstract
This part has too many details of methods, I suggest to add more of key results.
L29-30 Please add the specific number of reduced bacterial load, disease incidence, and severity.
L32-33 The same to last one. Please add specific data to make this sentence more persvasive.

Results
2.1.1 & Fig. 1 The authors need to check their data. It's strange why Famer's management (T1) was worse than controlled by only water (T0). The phage concentration of T2 and T3 did not vary a lot, it's hard to detect their significant difference.
2.2.1 & Fig. 2 The use of phage seems not better than water. Do I understand it right? Why use milk? Won't it cost too much for the production of kiwifruits?
2.2.2 & 2.2.3 it seems no data and figures for these two parts.

Author Response

Revisor 1.

Reviewer 1.  First of all thanks for contributing to improving this article!

Comments and Suggestions for Authors

Abstract
This part has too many details of methods, I suggest to add more of key results.

Response: Thanks for the suggestion. Some key results were added in red in the abstract.

L29-30 Please add the specific number of reduced bacterial load, disease incidence, and severity.

Response: Where specific bacterial load was reduced, the percentage of reduction was calculated and added to the text. Also, the percentage of reduced severity.

L32-33 The same to last one. Please add specific data to make this sentence more persuasive.

Ans: We introduce more specific data as suggested.

Results
2.1.1 & Fig. 1 The authors need to check their data. It's strange why Famer's management (T1) was worse than controlled by only water (T0).

Ans: This is not strange, considering that the farmer’s treatment was based on copper and antibiotics. We did not measure the resistance of Pss and Psa to copper and antibiotics, but other authors working in other countries have found copper and antibiotics resistance.

The phage concentration of T2 and T3 did not vary a lot, it's hard to detect their significant difference.

Ans: We cannot talk about the effective Psa and Pss load in each evaluation, but maybe if it was equal concentration, the efficacy of T3 treatment can be explained considering that contain ten times more phages than T2 treatment.

2.2.1 & Fig. 2 The use of phage seems not better than water. Do I understand it right? Why use milk? Won't it cost too much for the production of kiwifruits?

Ans: Figure 2. The treatment efficacy of was effective in three of the four trials done, at least in one evaluation per location. The negative effect on Peumo in the second season could be explained because some Pss were not controlled by the phage treatment. Milk was used just to demonstrate that treatment 4 that includes milk, the milk does not affect disease development. Even more, milk was not useful in the formulations for phage application (T4).

2.2.2 & 2.2.3 it seems no data and figures for these two parts.

Ans: Thanks for the observation. We add as suggested from reviewers, Figure 3 for bacterial load on first season of trials. Second season was not done because of COVID’s pandemia. And for Pseudomonas first identification approach we add LOPAT test based on Leliott [ 66] and molecular analysis. In the case of the six P. fluorescens only three isolates were later molecularly identified.

Reviewer 2 Report

Comments and Suggestions for Authors

Summary:

This study seeks to assess the effectiveness of bacteriophage cocktails against Pseudomonas syringae pv. actinidiae (Psa) for the purpose of disease control in kiwifruit orchards in a field setting. Previous work has utilized the selected phages in controlled settings and shown promising efficacy, but none have transferred the technology to a field setting. This study fills an important gap in applying bacteriophage sprays in the field to control disease caused by Psa and increase crop yield. The study was well designed, taking place over two seasons and in two different orchards in Chile, with control, farmer’s management, and phage formulations. However, the results presented are primarily based upon damage indexes, or % of leaves containing necrotic lesions, which may be biased depending on which leaves were selected for analysis. Raw data such as photographs of the necrotic areas compared between treatments are important include. Moreover, the effect of each treatment on the fruit production/ crop yield are necessary to demonstrate the effectiveness of the treatments. While these orchards have been known to have a presence of Psa and be naturally infected, the damage indices are not clearly linked to Psa infection. Although epiphytic bacterial populations were evaluated, and Psa and other Pseudomonas species were found, raw data indicating which leaves these were found in (treatments/ time points/ etc. ) and quantitative data on the levels of bacteria were lacking. Inclusion of these data is necessary to strengthen the paper and make a more convincing case for the effectiveness of the bacteriophage formulations. In the current state, not enough data is presented to support the effectiveness of bacteriophage treatment against Psa in the field.

Major comments:

  1. It would improve clarity of the paper to move Tables 1 and 2 into the results, as they contain information necessary to understand Figures 1 and 2 which appear earlier.
  2. The claim in line 394-395 “The results show that bacteriophage formulations consistently reduced the damage index (DI%), even surpassing copper treatments” isn’t accurate since the effects from the bacteriophage sprays were generally not significantly different from the control water sprays. In the first season especially, copper treatments seemed to cause increased plant damage in Linares, and in these cases it would be more appropriate to compare the results to the water only control.
  3. Inclusion of photographs of the leaves to support the Damage index data is important.
  4. What was the effect of each treatment on the kiwi fruit and the crop yield? This was not addressed in the paper.
  5. For the epiphytic bacterial populations, the methods state that plants were analyzed pre and post-treatment, but only bacterial populations pre-treatment were discussed in the results. What were the levels and composition of the bacterial populations before and after treatment? This is an important piece of data.
  6. No data for the pathogenicity tests or isolation and confirmation of the bacteria is shown. The statements are not supported by any micrographs, photos of infected leaves over time, etc.
  7. Section 2.1.1 of the Results was unclear and difficult to follow:
    1. Line 123-124: What is meant by “consistently low effect of farmer management”? Higher damage indices for farmer management than observed for other treatments?
    2. Lines 123-133: Figures A, B, C should be corrected to Figures 1A, 1B, 1C…
    3. Line 124: “During the second evaluation..” was not clear what this meant. Provide more context for the 30, 45, 90-day treatments and what you consider second or first evaluation.
    4. Line 132-133: “significant differences” is unclear- was there increased or decreased DI for these treatments?
    5. There is inconsistency in the epiphytic analysis- The results state this was performed in the first season, while the methods state this was performed in the second season.
  8. Data shown in Figures 1 and 2 could be more clearly explained.
    1. What do a, b, ab, c designations mean in terms of statistical significance? Please make this clear in the figure legends.
    2. What is the n? How many plants/ leaves are being analyzed in each graph?
  9. More information about the specific phages selected for the treatments would be helpful. Moreover, the pathogenicity tests combined with phage treatments would lend more support to the damage index being a result of infection with these specific bacterial isolates.

Minor comments:

  1. Lines 62-67: maintain consistency with formatting of Biovar vs. biovar and numbers.
  2. Line 149 and 156: Correct DDA to DAA
  3. Line 180: Spell out LOPAT and briefly summarize it.
  4. Line 190: change 92,3% to 92.3%.

Author Response

Reviewer 2.

First of  all thanks for your observations that improve our article!

Major comments:

1. It would improve clarity of the paper to move Tables 1 and 2 into the results, as they contain information necessary to understand Figures 1 and 2 which appear earlier.

Response: We consider that Tables 1 and 2 are more closely related to the methodological aspects rather than the Results section. However, the relevant information from these tables has been included in the figure legends, ensuring that Figures 1 and 2 can be fully understood without moving the tables.

2. The claim in line 394-395 “The results show that bacteriophage formulations consistently reduced the damage index (DI%), even surpassing copper treatments” isn’t accurate since the effects from the bacteriophage sprays were generally not significantly different from the control water sprays. In the first season especially, copper treatments seemed to cause increased plant damage in Linares, and in these cases it would be more appropriate to compare the results to the water only control.

Response: In the text we clarify more properly the results obtained in each trial. But in at least three of the four trials performed, in one or two evaluations the effect of Phage therapy was observed. The effect that copper and antibiotic treatment was consistent, something that we also obtained when another trial was performed under greenhouse conditions using copper as positive control. Here we include the obtained figure, and the article cite.  

Reference 42 in this article:

Flores, O.; Retamales, J.; Núñez, M.; León, M.; Salinas, P.; Besoain, X.; Yañez, C.; Bastías, R. Characterization of Bacteriophages against Pseudomonas syringae pv. actinidiae with Potential Use as Natural Antimicrobials in Kiwifruit Plants. Microorganisms 2020, 8, 974. https://doi.org/10.3390/microorganisms8070974

3. Inclusion of photographs of the leaves to support the Damage index data is important.

Response: Thanks for the information. This was included in the supplementary data (Figure 1).

4. What was the effect of each treatment on the kiwi fruit and the crop yield? This was not addressed in the paper.

Response: Because of the pandemia we could not measure yield in April 2020 and April 2021. But it is important to note that Psa and Pss does not affect kiwi fruits, only in an indirect way.

5. For the epiphytic bacterial populations, the methods state that plants were analyzed pre and post-treatment, but only bacterial populations pre-treatment were discussed in the results. What were the levels and composition of the bacterial populations before and after treatment?

Response: Thanks for your observation. Yes, we measure bacterial populations, but only in the first year. Now we are including new data, Figure 1 and Tables 1 and 2, concerned with the bacterial epiphytic load and the identification of representative bacteria. There is one evaluation done in Peumo and Linares where the bacterial load was reduced associated with a decrease of damage associated with phage therapy. This result was included in red in the text.

6. No data for the pathogenicity tests or isolation and confirmation of the bacteria is shown. The statements are not supported by any micrographs, photos of infected leaves over time, etc.

Response: Thanks for the observation. We include the results obtained in pathogenicity test (see Supplementary Data, Table 1).

7. Section 2.1.1 of the Results was unclear and difficult to follow:

         1. Line 123-124: What is meant by “consistently low effect of farmer management”? Higher damage indices for farmer management than observed for other treatments?

Response: Yes, that was the original idea. We redact a new paragraph to be more clear.

         2. Lines 123-133: Figures A, B, C should be corrected to Figures 1A, 1B, 1C…

Response: All these data were corrected. We also include in the figures first Peumo and later Linares, to be consistent with the methodology.

         3. Line 124: “During the second evaluation..” was not clear what this meant. Provide more context for the 30, 45, 90-day treatments and what you consider second or first evaluation.

Response: We correct the text as suggested.

          4. Line 132-133: “significant differences” is unclear- was there increased or decreased DI for these treatments?

Response: This observation was corrected in the text.

          5. There is inconsistency in the epiphytic analysis- The results state this was performed in the first season, while the methods state this was performed in the second season.

Response: We correct this inconsistence. The measure was done in the first season.

9. Data shown in Figures 1 and 2 could be more clearly explained.

             1. What do a, b, ab, c designations mean in terms of statistical significance? Please make this clear in the figure legends.

Response: All these observations were corrected.

             2. What is the n? How many plants/ leaves are being analyzed in each graph?

Response: The n= 10 leaves per treatment and block.

10. More information about the specific phages selected for the treatments would be helpful. Moreover, the pathogenicity tests combined with phage treatments would lend more support to the damage index being a result of infection with these specific bacterial isolates.

Response: This information was included in Methodology chapter as Table 3.

Minor comments:

1. Lines 62-67: maintain consistency with formatting of Biovar vs. biovar and numbers.

Response: This correction was done.

2. Line 149 and 156: Correct DDA to DAA

Response: Thanks for the information. This correction was done.

3. Line 180: Spell out LOPAT and briefly summarize it.

Response: This correction was done

4. Line 190: change 92,3% to 92.3%.

Response: This correction was done.

Reviewer 3 Report

Comments and Suggestions for Authors

The manuscript entitled: „Bacteriophages Biocontrol of Kiwifruit Bacterial Canker caused
by Pseudomonas syringae pv. actinidiae (Psa) in two seasons under field conditions” describes the biocontrol potential of bacteriophage mixtures against Psa in different seasons and localities compared to farmers' traditional management.

Generally, my main concern is regarding the experimental design and methods for the analysis of bacterial epiphyte populations. More detailed information in the methods and results sections are required, as well as additional experiments for the pathogenicity test.

More specifically, in the methods section, PCR conditions for all specific primers used should be included. The information on positive and negative controls included in LOPAT tests should be provided. When performing pathogenicity test, I strongly recommend the authors to do LOPAT tests with re-isolated bacteria, as well as confirmation with specific primers.

Results regarding the same section should also be explained in more detail. The authors should be more specific regarding the mentioned identification based on sequencing and why 17 isolates were selected?  Results obtained with controls for LOPAT tests should also be included.

Were epiphytic bacterial populations isolated from both localities? Please add which strains were isolated from which locality. Could the different effects of the same treatments in Linares and Peumo be explained?

Additionally, I recommend adding a figure for the pathogenicity test.

Accordingly, the authors should expand the discussion section. Other examples of bacteriophage-mediated biocontrol of P. syringae should be added and compared to the results obtained in this study.

Do different effects of treatments based on time of sampling tell us something about potential applicability in the future?

Minor:

Line 332 – Please change „second“ to „first“

Line 333 and 334 – Please rephrase this sentence.

Author Response

Reviewer 3.

Reviewer 3.

Dear reviewer, thanks for your observations that improve our article!

Comments and Suggestions for Authors

The manuscript entitled: „Bacteriophages Biocontrol of Kiwifruit Bacterial Canker caused
by Pseudomonas syringae pv. actinidiae (Psa) in two seasons under field conditions” describes the biocontrol potential of bacteriophage mixtures against Psa in different seasons and localities compared to farmers' traditional management.

Generally, my main concern is regarding the experimental design and methods for the analysis of bacterial epiphyte populations. More detailed information in the methods and results sections are required, as well as additional experiments for the pathogenicity test.

Response: Thanks for the information. We include the analysis of bacterial epiphyte populations, please see Table 1, Table 2 and Suplementary Data Table 1.

More specifically, in the methods section, PCR conditions for all specific primers used should be included. The information on positive and negative controls included in LOPAT tests should be provided. When performing pathogenicity test, I strongly recommend the authors to do LOPAT tests with re-isolated bacteria, as well as confirmation with specific primers.

Response: Thanks for the observations. PCR conditions were included in methodology, and results included in Table 2. For LOPAT test this in something that we generally do in our lab and so the results are always clear. Only in Tobaco Hipersensitive reaction we perform the test including positive and negative controls. In the case of pathogenicity test we did not analyze LOPAT again, Pss and Psa are already confirmed by other authors that both species and patovars can cause foliar symptoms.

Results regarding the same section should also be explained in more detail. The authors should be more specific regarding the mentioned identification based on sequencing and why 17 isolates were selected?  Results obtained with controls for LOPAT tests should also be included.

Response: All this information was included in Tables 1 and 2, and in Supplementary data.

Were epiphytic bacterial populations isolated from both localities? Please add which strains were isolated from which locality. Could the different effects of the same treatments in Linares and Peumo be explained?

Response: Considering new information included in Figure 3, we can observe in general that bacterial load thus does not differ in both areas, Peumo and Linares.

Additionally, I recommend adding a figure for the pathogenicity test.

Response: We include this information in Supplementary Data Table 1.

Accordingly, the authors should expand the discussion section. Other examples of bacteriophage-mediated biocontrol of P. syringae should be added and compared to the results obtained in this study.

Response: The discussion session was expanded, we include other trials done on kiwifruit plants.

Do different effects of treatments based on time of sampling tell us something about potential applicability in the future?

Response: In this context considering in vitro, in vivo and field results, this can applied in the field. It could be better to include in the formula phages that contribute to control Pss bacteria also, considering that according to pathogenicity test probable Pss play a rol in kiwi leaves foliar symptoms.

Minor:

Line 332 – Please change „second“ to „first“

Response: Correction done.

Line 333 and 334 – Please rephrase this sentence.

Response: This sentence was rephrased.

Round 2

Reviewer 1 Report

Comments and Suggestions for Authors

The authors answered all my questions. I suggest to merge their answers into the Discussion, because I think other readers will also bring up these questions.

Author Response

Open Review

Comments and Suggestions for Authors

The authors answered all my questions. I suggest to merge their answers into the Discussion, because I think other readers will also bring up these questions.

First, we would like to thank you for your valuable feedback, which has significantly contributed to improving the quality and clarity of our manuscript.

Thanks for the suggestion. In the discussion, these aspects were included in Lines 267-271 and Lines 281-288.

Reviewer 2 Report

Comments and Suggestions for Authors

Thank you for including the additional figures and supplementary data, which very much improves the paper. Showing the parameters for how damage index was determined on infected leaves helped lend support to the figures. The additional LOPAT data and bacterial loads were also very helpful to show that diseased leaves were indeed a result of the presence of Psa and that the phage treatments significantly reduced overall bacterial load. The following comments are minor suggestions for improving grammar and clarity in the final published version:

Figure 3: Please change “UFC” to “CFU” on the y-axes for each graph

Line 134: Change “significatively” to “significantly”

Table 1 caption: Revise “pseudomonas fluorescent group” to “Pseudomonas fluorescens group”

Line 201-211: The following paragraphs should be revised for clarity/ grammar:

 “Considering epiphytic bacteria load undoubtedly was high at the beginning of the measurements and in the control treatment (T0) (Figure 3 A, D). The bacterial load decreases up to 43% in the evaluation done 60 days after treatment, and this difference resulted in a decrease in foliar damage in the first season of evaluation done in Peumo (Figure 1 B,C).

In the recovery of representative bacteria from epiphytic population (Table 1, 2) the presence of P. fluorecens and other Pseudomonas species (Table 1) and other geniuses (Table 2), could exert a natural control of Psa and Pss bacteria, and probably most of them are resistant to copper or antibiotics treatments. This issue should be evaluated and measured in the future.

Suggestion for revision:

“Epiphytic bacteria load was high in the control treatment (T0) group and in the early stages of treatment, ranging from 10^6 to 10^8 CFU/ mL (Figure 3 A, D). The bacterial load decreased by up to 43% after 60 days of treatment with phage formulations 1 and 2 (Figure 3C), corresponding with a decrease in foliar damage in the first season of evaluation done in Peumo (Figure 1 B,C).

Representative bacteria were recovered from epiphytic populations, including P. fluorescens, other Pseudomonas species, and other genuses (Tables 1 and 2). These organisms could exert a natural control of Psa and Pss bacteria, and may exhibit resistance to copper or antibiotics treatments. The effect of these bacterial populations should be evaluated in the future.”

Additionally, with the new data added, the results section switches between Figure 3 and Tables 1 and 2. This could be revised to flow better, discussing first the bacterial loads and then going into the LOPAT analysis.

Line 279: Change “probable” to “probably”

Author Response

Thank you for including the additional figures and supplementary data, which very much improves the paper. Showing the parameters for how damage index was determined on infected leaves helped lend support to the figures. The additional LOPAT data and bacterial loads were also very helpful to show that diseased leaves were indeed a result of the presence of Psa and that the phage treatments significantly reduced overall bacterial load. The following comments are minor suggestions for improving grammar and clarity in the final published version:

First of all, thanks for your contribution that has improved our manuscript.

Figure 3: Please change “UFC” to “CFU” on the y-axes for each graph

Response: We have corrected Figure 3 by changing “UFC” to “CFU” on the y-axis of all graphs.

Line 134: Change “significatively” to “significantly”

Response: We have corrected the text by changing “significatively” to “significantly” in the revised manuscript.

Table 1 caption: Revise “pseudomonas fluorescent group” to “Pseudomonas fluorescens group” 

Response: In this case, however, we decided to keep the caption as “Pseudomonas fluorescent group,” since the LOPAT scheme is used not only to identify bacteria belonging to the Pseudomonas fluorescens species, but also other fluorescent bacteria of the genus Pseudomonas, such as P. syringae, in our study. So, to be more precise we decided to describe as “Group of fluorescent Pseudomonas”.

Line 201-211: The following paragraphs should be revised for clarity/ grammar:

 “Considering epiphytic bacteria load undoubtedly was high at the beginning of the measurements and in the control treatment (T0) (Figure 3 A, D). The bacterial load decreases up to 43% in the evaluation done 60 days after treatment, and this difference resulted in a decrease in foliar damage in the first season of evaluation done in Peumo (Figure 1 B,C).

In the recovery of representative bacteria from epiphytic population (Table 1, 2) the presence of P. fluorecens and other Pseudomonas species (Table 1) and other geniuses (Table 2), could exert a natural control of Psa and Pss bacteria, and probably most of them are resistant to copper or antibiotics treatments. This issue should be evaluated and measured in the future.

Suggestion for revision:

“Epiphytic bacteria load was high in the control treatment (T0) group and in the early stages of treatment, ranging from 106 to 108 CFU/ mL (Figure 3 A, D). The bacterial load decreased by up to 43% after 60 days of treatment with phage formulations 1 and 2 (Figure 3C), corresponding with a decrease in foliar damage in the first season of evaluation done in Peumo (Figure 1 B,C).

Representative bacteria were recovered from epiphytic populations, including P. fluorescens, other Pseudomonas species, and other genuses (Tables 1 and 2). These organisms could exert a natural control of Psa and Pss bacteria, and may exhibit resistance to copper or antibiotics treatments. The effect of these bacterial populations should be evaluated in the future.”

Response: We insert, as suggested, your paragraph with minor changes in Lines 197-213.

Additionally, with the new data added, the results section switches between Figure 3 and Tables 1 and 2. This could be revised to flow better, discussing first the bacterial loads and then going into the LOPAT analysis.

Response: Following your suggestion, we have reorganized the section to ensure a more coherent structure.

Line 279: Change “probable” to “probably”

Response: We thank the reviewer for the suggestion. The term “probable” has been corrected to “probably” in the revised manuscript.

Reviewer 3 Report

Comments and Suggestions for Authors

Minor comments:

Line 135 – Please change “both” to “all three”

Line 213 – This figure is not about LOPAT or molecular identification

Line 228 – Please change “gyrB” to “syrB”

In table 2 please correct typing mistakes

In the discussion, please do not refer to tables and supplementary 

Author Response

Reviewer 3

We sincerely appreciate your detailed review and insightful comments, which have been instrumental in refining and strengthening our manuscript.

Line 135 – Please change “both” to “all three”

Response: We have replaced “both” with “all three” in the revised version of the manuscript to reflect the number of components accurately.

Line 213 – This figure is not about LOPAT or molecular identification

Response: We have removed the reference to LOPAT and molecular identification, as this figure is not related to those analyses.

Line 228 – Please change “gyrB” to “syrB”

Response: We have corrected the text by changing “gyrB” to “syrB” in the revised version of the manuscript.

In table 2, please correct typing mistakes

Response: All typing mistakes in Table 2 have been carefully revised and corrected in the updated version of the manuscript.

In the discussion, please do not refer to tables and supplementary 

Response: We have removed all comments referring to tables and supplementary material from the Discussion section.